# A Silicon Sub-Bandgap Near-Infrared Photodetector with High Detectivity Based on Textured Si/Au Nanoparticle Schottky Junctions Covered with Graphene Film

**DOI:** 10.3390/s23136184

**Published:** 2023-07-06

**Authors:** Xiyuan Dai, Li Wu, Kaixin Liu, Fengyang Ma, Yanru Yang, Liang Yu, Jian Sun, Ming Lu

**Affiliations:** 1Department of Optical Science and Engineering, and Shanghai Ultra-Precision Optical Manufacturing Engineering Center, Fudan University, Shanghai 200433, China; 20110720001@fudan.edu.cn (X.D.); 21110720018@m.fudan.edu.cn (L.W.); 21210720009@m.fudan.edu.cn (K.L.); 22110720009@m.fudan.edu.cn (F.M.); 22210720005@m.fudan.edu.cn (Y.Y.); 21210720284@m.fudan.edu.cn (L.Y.); 2Yiwu Research Institute, Fudan University, Yiwu 322000, China

**Keywords:** Si sub-bandgap near-infrared photodetector, Schottky junction, internal photoemission, graphene

## Abstract

We present a straightforward approach to develop a high-detectivity silicon (Si) sub-bandgap near-infrared (NIR) photodetector (PD) based on textured Si/Au nanoparticle (NP) Schottky junctions coated with graphene film. This is a photovoltaic-type PD that operates at 0 V bias. The texturing of Si is to trap light for NIR absorption enhancement, and Schottky junctions facilitate sub-bandgap NIR absorption and internal photoemission. Both Au NPs and the texturing of Si were made in self-organized processes. Graphene offers additional pathways for hot electron transport and to increase photocurrent. Under 1319 nm illumination at room temperature, a responsivity of 3.9 mA/W and detectivity of 7.2 × 10^10^ cm × (Hz)^1/2^/W were obtained. Additionally, at −60 °C, the detectivity increased to 1.5 × 10^11^ cm × (Hz)^1/2^/W, with the dark current density reduced and responsivity unchanged. The result of this work demonstrates a facile method to create high-performance Si sub-bandgap NIR PDs for promising applications at ambient temperatures.

## 1. Introduction

Photodetectors (PDs), which convert light into electrical signals, play a crucial role in various fields such as data transmission [1], night vision imaging [2], biomedical testing [3], and automotive radar system [4]. Silicon (Si) PDs are compatible with contemporary complementary metal-oxide-semiconductor (CMOS) technology, offering cost-effectiveness and technical availability for electronic and photonic integration [5,6]. At present, Si PDs are excellent detectors at visible wavelengths. However, Si PDs that can operate at sub-bandgap near-infrared (NIR) wavelengths, or Si sub-bandgap NIR PDs, have been a big challenge, since NIR light with photon energy lower than the Si bandgap width of 1.1 eV, or with a wavelength longer than 1100 nm, is transparent to Si. To address this issue, three mechanisms have been proposed so far, which are the internal photoemission effect (IPE), surface or bulk defect-mediated absorption (SDA or BDA), and two-photon absorption (TPA) [5,6]. Among them, IPE-based PDs exhibit relatively good controllability in fabrication, but their responsivities, and especially detectivities, need much improvement for practical applications. For this purpose, various efforts have been made to enhance the NIR absorption [7,8,9], strengthen the NIR-induced carrier transport and separation [10,11], and reduce dark current [12], and novel approaches for creating Si sub-bandgap NIR PDs with superior performance and low fabrication cost are still in demand. In this work, we propose a facile method to make a Si sub-bandgap NIR PD with high detectivity, which works at ambient temperatures. As the primary structure of the Si PD, Si/Au nanoparticle (NP) Schottky junctions were prepared on textured Si, with both Au NPs and texturing of Si made through self-organized processes. Furthermore, considering that graphene (Gr) has a longer hot electron relaxation time and higher hot carrier temperature [13], Gr was introduced for higher hot carrier emission efficiency [14,15,16]. To achieve low dark current for high detectivity, the photovoltaic type of PD operating at 0 V bias was adopted for the proposed NIR PD based on textured Si/Au NPs/Gr structure in this work. Under 1319 nm illumination, a responsivity of 3.9 mA/W and detectivity of 7.2 × 10^10^ cm × (Hz)^1/2^/W were obtained at room temperature. At −60 °C, the detectivity increased to 1.5 × 10^11^ cm × (Hz)^1/2^/W due to a further reduction in dark current.

## 2. Experiment

### 2.1. Preparation of Textured Si Substrate

N-type Si(100) wafer (Rdmicro, Suzhou, China, 1~10 Ω∙cm, two sides polished, and 200 ± 10 μm thick) was chosen as the substrate. The Si wafer was ultrasonically cleaned in acetone (Dahe Chemicals, Shanghai, China), ethanol (Titan, Shanghai, China), and deionized water in sequence, and then dried with nitrogen. To fabricate textured Si with pyramid structure, the Si wafer was etched in a solution of NaOH (Sinopharm Chemical Reagent Co., Ltd., Shanghai, China, 2 wt%), Na_2_SiO_3_ (Aladdin, Shanghai, China, 2 wt%), and isopropyl alcohol (Sinopharm Chemical Reagent Co., Ltd., Shanghai, China, 7 vol%) at 80 °C for 25 min. Textured Si could enhance the light absorption through light trapping and simultaneously facilitate the subsequent formation of Si/Au NPs Schottky junctions [17,18]. The textured Si was then rinsed in deionized water and cleaned in a solution of H_2_O:H_2_O_2_ (Sinopharm Chemical Reagent Co., Ltd., Shanghai, China, 30%):NH_3_ (Sinopharm Chemical Reagent Co., Ltd., Shanghai, China, 25%) = 6:1:1 at 70 °C for 20 min and H_2_O:H_2_O_2_(30%):HCl (Sinopharm Chemical Reagent Co., Ltd., Shanghai, China, 36%) = 6:1:1 at 70 °C for 10 min in sequence. Finally, the wafer was dipped in dilute HF (Sinopharm Chemical Reagent Co., Ltd., Shanghai, China, 1%) for 1 min to remove the naturally formed surface Si oxide.

### 2.2. Preparation of PD Device

Au (ZhongNuo Advanced Material Technology Co., Ltd., Beijing, China) thin film was deposited on the front side of the textured Si at a rate of 0.05 nm/s by resistance heating in a vacuum chamber at a base pressure lower than 6 × 10^−4^ Pa. Subsequently, it was annealed in a forming gas of hydrogen and nitrogen (H_2_:N_2_ = 5%:95% in volume) at 450 °C for 30 min to form Au NPs, and, thus, Si/Au NPs Schottky junctions. To deposit Gr film, the Gr suspension (XFNano, Nanjing, China, 0.5 mg/mL) mixed with alcohol (1:3 volume ratio) was dropped using a micropipette and spin-coated onto the surface of Au-NP-incorporated textured Si (50 μL, 5000 rpm, and 10 s). The Gr was then firmly adhered to the Au NPs and the exposed Si surface. It could provide bypass pathways to transport hot electrons produced in Au NPs to the conduction band of Si through Si/Gr contact [19,20], in addition to the Si/Au NP contact pathway. A 100.0 nm thick indium tin oxide (ITO, ZhongNuo Advanced Material Technology Co., Ltd., Beijing, China) layer was deposited as the front electrode. A 20.0 nm thick SiO_2_ (ZhongNuo Advanced Material Technology Co., Ltd., Beijing, China) layer was made on the backside of the textured Si by electron beam evaporation for surface passivation, and a 1.0 μm thick Al (ZhongNuo Advanced Material Technology Co., Ltd., Beijing, China) layer was grown via resistance heating as the rear electrode. ITO was chosen to be the front electrode because it is a typical transparent and conductive oxide thin film. For the rear electrode, we chose Al because there is a standard process to form Al-Si ohmic contact. The SiO_2_ passivation layer at the rear could serve to saturate dangling bonds on the Si surface, thereby reducing the trapping of photoinduced charges by these defects [21]. The thickness of SiO_2_ was optimized to maximize the passivation effect and avoid the influence of larger resistance. Finally, thermal annealing was conducted at 450 °C in nitrogen for 5 min to form ohmic contacts. The surface area of the PD was 1 cm × 1 cm. Figure 1 presents a schematic illustration of the textured Si/Au NPs/Gr NIR PD.

### 2.3. Characterization

The surface morphology of the Schottky junction was measured with scanning electron microscopy (SEM, Philips XL30, Philips, Amsterdam, The Netherlands). The thickness of the Gr film was measured using an atomic force microscope (AFM, Bruker Dimension Icon, Mannheim, Germany). The absorption spectra of the Schottky junction were obtained using a vis-NIR spectrometer (Ideaoptics, NIR2500, Shanghai, China) with an integrating sphere. The Raman spectrum of the Gr film was tested on an instrument of Renishaw InVia (Wotton-under-Edge, UK) before and after contact with Au. The current–voltage (I-V) characteristics of the PD were measured at room temperature under darkness or NIR light illumination, using a source meter (Keithley, SMU2400, Cleveland, OH, USA). The NIR light source was a 1319 nm laser diode (CNI laser, MIL-H-1319, Changchun, China). When measuring the I-V characteristics under light conditions, the front side of the PD faced a 1319 nm light beam with illumination power of 0.1 W/cm^2^. The external quantum efficiency (EQE) of the PD was measured with a QE/IPCE system of Oriel/Newport (Irvine, CA, USA). A photoresponse measurement of the PD under low temperature was conducted at −60 °C in an in-house cryochamber. The surface potential of Schottky junction was tested with a Kelvin probe force microscope (KPFM) equipped on the AFM. To obtain the time response of the PD, the rise and fall time at 0 V bias were tested with an oscilloscope (Siglent, SDS1202X-E, Shenzhen, China) when the device was illuminated by the 1319 nm light modulated through a chopper (Daheng Optics, GCI-15, Shanghai, China).

## 3. Results and Discussion

Figure 2 displays the surface and cross-sectional SEM images of the textured Si/Au NPs coated with Gr. The Gr was found to adhere tightly to both Au NPs and the remaining Si surface area. The thickness of the Gr film spin-coated on planar Si ranged from 5 to 10 nm (Appendix A). Given that the thickness of a single layer of Gr is approximately 1.0 nm [22], the spin-coated Gr on the PD device could be considered multilayered. The textured Si surface exhibited micrometer-sized pyramid-like structures with an average base width of 8.0 ± 5.1 μm and an average height of 5.6 ± 2.8 μm, as shown in Figure 2b. The deposited Au film had an apparent thickness of 25.0 nm. After annealing, Au NPs were formed on the textured Si surface, with sizes of 40.1 ± 25.9 nm. On the other hand, a small number of larger Au NPs, approximately 100 nm in size, appeared on the ridges of the Si pyramids, which could be more favorable for Au NP nucleation [18].

The absorption spectra of the textured Si, textured Si/Au NPs, and textured Si/Au NPs/Gr in the NIR wavelength range of 1200–1800 nm were measured, as shown in Figure 3. For comparison, the absorption of planar Si and planar Si/Au NPs was also plotted. The average NIR absorption of the textured Si was approximately 25%, which was attributed to the antireflection effect and doped impurity absorption. This higher NIR absorption compared to the planar Si was in accordance with the theoretical calculations of Si nanocone arrays [23]. Following the formation of Au NPs, the NIR absorption of the textured Si/Au NP sample increased significantly to an average of about 70%. This enhanced absorption stemmed from the Au NPs, where localized surface plasmons (LSPs) were induced by incident NIR light, generating energetic hot electrons [7,18,24,25]. These hot electrons were primarily responsible for photodetection by forming a photoinduced current through the IPE process [26,27]. Au NPs with identical sizes distributed on the Si substrate would show a resonance absorption peak due to LSPs [8]. In a fabricated sample, Au NPs with various sizes would result in a broadband absorption spectrum [8], as demonstrated in Figure 3. According to the FDTD simulation results in reference [24], for textured Si/Au NPs’ structure, there was higher optical absorption in the NIR region and a larger enhancement of the localized electric field compared with planar Si/Au NPs’ structure. The experimental data in Figure 3 also indicate higher NIR absorption for the textured Si/Au NPs, consistent with the FDTD simulation. This phenomenon could be explained by the tilted interface of the textured Si, which reflected light to the adjacent pyramids, increased the optical path, and strengthened the interaction between light and Au NPs [24]. After introducing the Gr film, the absorption of the textured Si/Au NPs/Gr was further improved slightly by approximately 1%, since Gr itself can also absorb NIR light [28]. Since the broadband NIR absorption remained nearly unaffected, it is inferred that the introduction of Gr does not disturb the LSPs on Au NPs; only the resonance absorption peak might experience a slight red shift by tens of nanometers [29].

Figure 4 presents Raman spectra of the planar Si/Gr and planar Si/Au NPs/Gr samples. Both Raman spectra exhibit the characteristic D, G, and 2D peaks of Gr, which are typically situated at approximately 1350 cm^−1^, 1580 cm^−1^, and 2700 cm^−1^, respectively [30,31], indicating that Gr did exist on the Si/Au NP junctions. It was also observed that the presence of Au NPs led to a notable increase in the intensity of the Gr Raman peaks. This is consistent with the result of creating a Gr-Au hybrid structure for high-performance surface-enhanced Raman scattering [32], and is ascribed to the enhanced electromagnetic field induced by the LSP resonance of Au NPs [32,33].

Figure 5a,b display the I-V characteristics of the textured Si/Au NPs and textured Si/Au NPs/Gr PDs under dark and 1319 nm light illumination conditions at room temperature. Responsivity (*R*) is a crucial figure of merit for PDs as it measures the electrical response to light, and is defined as [34]
(1)$$R=\frac{I_{p}}{P_{in}}=\frac{I_{light}-I_{dark}}{P_{in}},$$
where *I_p_* and *P_in_* are the photoinduced current and incident light power; *I_light_* and *I_dark_* are the measured current under light illumination and darkness, respectively. Based on this equation, *R* was calculated to be 3.4 mA/W for the textured Si/Au NP device and 3.9 mA/W for the textured Si/Au NPs/Gr one, both working at 0 V bias. It is worth pointing out that the strong plasmonic effect of Au NPs was a critical factor leading to good responsivities [35,36,37]. In our previously reported work, the device fabricated with Au thin film showed responsivities lower than 1 mA/W [18]. Compared with the case of Au thin film, the localized surface plasmons generated by Au NPs could enhance the NIR absorption and localized electric field, which increased the number and kinetic energy of hot electrons, thus enhancing the photocurrent [24,38]. Moreover, the presence of Gr induced a higher photocurrent. This is because the Gr film offers an additional pathway for hot electron extraction, increasing the IPE efficiency. It is noted that the dark current under reverse bias was reduced when introducing the Gr film, as shown in Figure 5a. The saturation current density of the Schottky junction at reverse bias can be expressed as [39]
(2)$$J_{S}=A^{*}T^{2}\exp(-\frac{e\phi_{b}}{kT}),$$
where *A** is the Richardson constant and $\phi_{b}$ is the Schottky barrier height. Thus, the lower saturation current after adding the Gr is due to the increment in the Schottky barrier height ($\phi_{b}$). By fitting these typical I-V curves to the thermionic emission equation [39], the $\phi_{b}$ was obtained as 0.59 eV and 0.61 eV for the Si/Au NP structure device and the Si/Au NPs/Gr one, respectively. For a Schottky diode, the open circuit voltage (V_OC_) is linearly proportional to the Schottky barrier height [40]. From Figure 5b, the V_OC_ increased by 0.02 V, which is consistent with the barrier height improvement between the two devices. The tested EQE of the textured Si/Au NPs/Gr PD at 1100–1600 nm is drawn in Figure 5c. Because of the NIR absorption and realization of IPE, the EQE at 1200–1400 nm remained larger than zero, with an EQE value of 0.1% at 1319 nm. Considering the definition of EQE, it could be calculated using
(3)$$\mathrm{EQE}=\frac{I_{p}/e}{P_{in}/h\upsilon },$$
where *hυ* is the photon energy. By using the data in I-V characteristics, the EQE at 1319 nm could be calculated as ~0.3%, which was close to the measured value.

Specific detectivity (*D**), on the other hand, is another important figure of merit, which measures detector sensitivity and determines the ability to distinguish weak light signals from noise. This parameter is particularly significant for NIR testing and imaging, as a higher *D** indicates a more sensitive PD. Assuming that shot noise from dark current is the major contributor to the total noise, *D** can be expressed as [34]
(4)$$D^{*}=\frac{R}{{\left(2q\cdot J_{dark}\right)}^{1/2}},$$
where *q* is the unit charge and *J_dark_* is the dark current density. From Equation (4), *D** was deduced to be 4.1 × 10^10^ and 7.2 × 10^10^ cm × (Hz)^1/2^/W for the textured Si/Au NPs without and with Gr, respectively, operating at 0 V bias. The higher detectivity of the textured Si/Au NPs/Gr PD could be attributed to the fairly good responsivity and lower dark current at 0 V bias. In order to further reduce the dark current and elevate the *D**, we tested the photoresponse of the Si/Au NPs/Gr PD device working at a lower ambient temperature (Appendix A). Compared with the performance at room temperature, the PD operating at −60 °C had a further lower dark current of 2 nA, and the *D** increased further to 1.5 × 10^11^ cm × (Hz)^1/2^/W. In Table 1, the responsivity, detectivity, and dark current density are listed for our Si sub-bandgap NIR PDs, compared to the other results of PDs with testing conditions at 0 V bias and room temperature [24,41,42,43]. The responsivities of the listed Si/Au structure PDs were in the same magnitude, while our proposed PD evidently had a lower *J_dark_*, leading to higher detectivity. The lower *J_dark_* could be attributed to the high barrier height [25] and optimized surface passivation [44]. In comparison to PDs with other structures working in the NIR wavelength range, the responsivities in this work were relatively lower, which could be explained by the difference in light absorption materials.

To investigate the interfacial built-in electric field of the Schottky junction, the surface potentials of planar Si/Au NPs and planar Si/Gr were acquired using a KPFM, as shown in Figure 6a,b, respectively. The tested average surface potentials of the Au NPs and Gr were −420 mV and −130 mV, and the potential of the Si region was 440 mV. The potential value of Au was lower than Si by 860 mV, which conformed well with the larger work function of Au (~5.1 eV) than that of n-Si (~4.3 eV) by ~0.8 eV [24]. The work function of Gr was calculated as ~4.8 eV according to the relatively higher potential compared with Au [45], consistent with the value (4.8~5.0 eV) in reference [46]. Figure 6c,d illustrate the energy band diagram of the Si/Au NPs/Gr Schottky junction, and the band alignment of each component in the PD device, respectively. Due to the work function difference between Gr and n-Si verified by the KPFM results, a Schottky junction also formed at their interface [42,47]. Since Gr was p-type-doped after contact with Au [33], the built-in electric field between the Gr and Si aligned with the one created by the Au/n-Si interface. Hot electrons generated in Au NPs can transfer into the conduction band of a Gr sheet directly because Gr has no bandgap [16]. Consequently, hot electrons could enter the conduction band of Si through the two Schottky junction pathways. In this case, hot electrons in Au NPs with lateral momentum can transfer to Gr first and then enter Si, resulting in an improved photocurrent and, thus, enhanced *R* and *D**, as displayed in Table 1.

Figure 7 shows the temporal response of the textured Si/Au NPs/Gr device at room temperature under 0 V bias. With the chopper operating at 3000 Hz, the photoresponse demonstrated that the PD device can follow a modulated light with a frequency of at least 3000 Hz. The estimated rise time and fall time were 31 μs and 55 μs, respectively. These results manifested the high response speed of the Si/Au NPs/Gr device, compared with the Si/Gr Schottky junction PDs [14,42]. Once LSP resonances are excited on Au NPs, they quickly decay into electron–hole pairs through Landau damping at the femtosecond scale [48]. Therefore, the speed of a PD is mainly limited by the carrier drifting time from NPs to electrodes and the resistance capacitance (RC) constant [10].

## 4. Conclusions

In summary, we demonstrated a facile method to make a high-performance Si sub-bandgap NIR Schottky junction PD, i.e., a textured Si/Au NP/Gr PD. The Au NPs and the texturing were all made in a self-organized manner, and the photovoltaic mode of PD was adopted. A responsivity of 3.9 mA/W and specific detectivity of 7.2 × 10^10^ cm × (Hz)^1/2^/W were achieved at room temperature. The detectivity increased to 1.5 × 10^11^ cm × (Hz)^1/2^/W, with the responsivity unchanged at −60 °C. By further increasing the NIR absorbance and transport of hot electrons via optimizing the Si texturing condition and band bending of the Schottky junction, the performance of the PD can be further improved. This work provides a practical route for developing a low-cost and highly efficient Si sub-bandgap NIR Schottky junction PD.

## Figures and Tables

**Figure 1 sensors-23-06184-f001:**
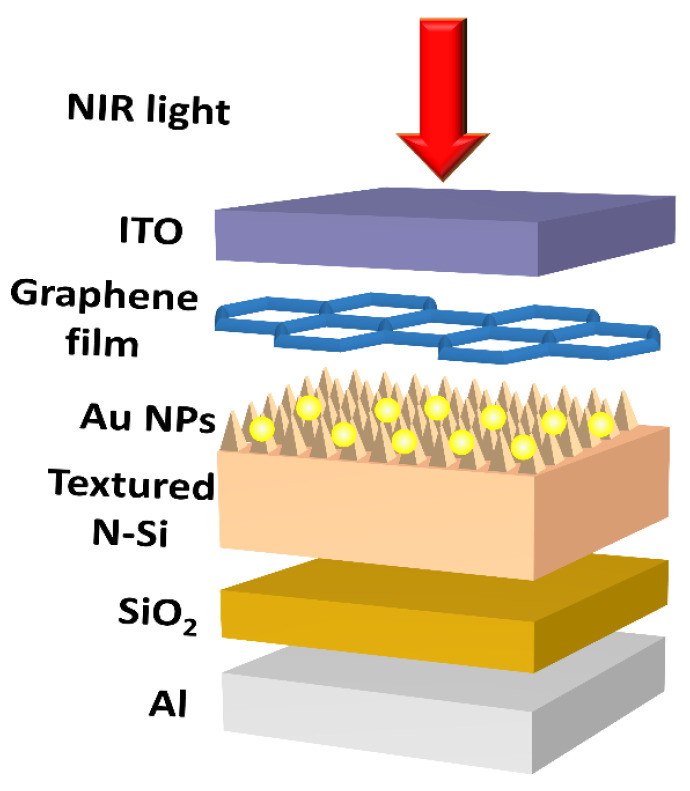
Schematic illustration of Si/Au NPs/Gr Schottky junction near-infrared PD.

**Figure 2 sensors-23-06184-f002:**
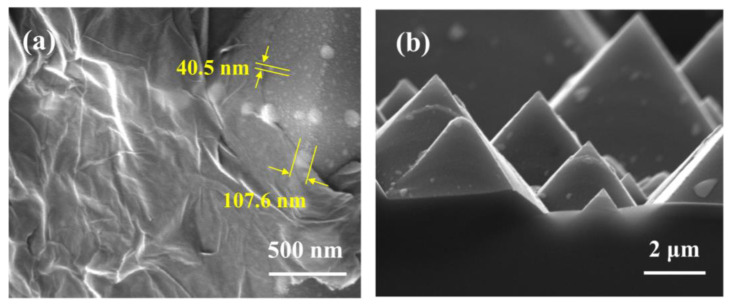
SEM surface (**a**) and cross-sectional (**b**) image of textured Si/Au NPs/Gr.

**Figure 3 sensors-23-06184-f003:**
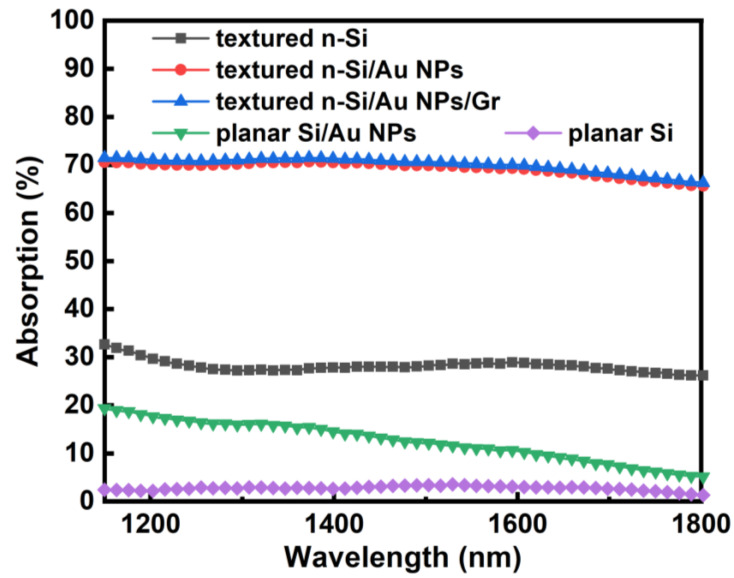
NIR absorption spectra of planar Si, textured Si, planar Si/Au NPs, textured Si/Au NPs, and textured Si/Au NPs/Gr.

**Figure 4 sensors-23-06184-f004:**
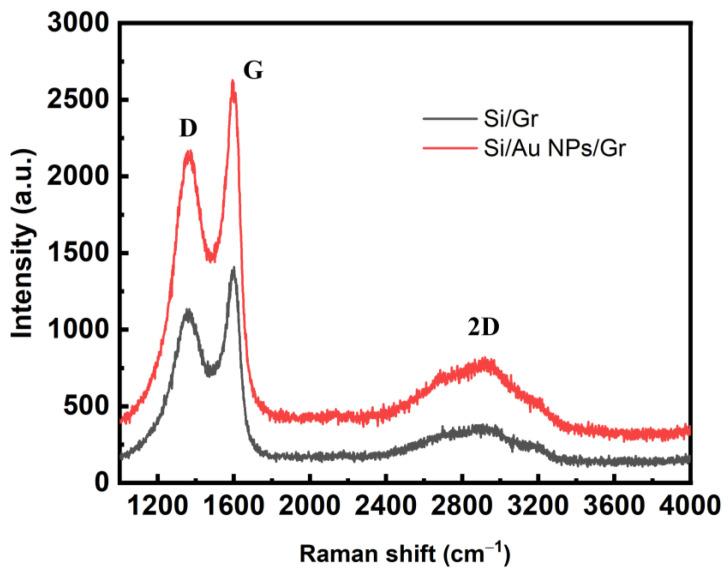
Raman spectra of planar Si/Gr and planar Si/Au NPs/Gr.

**Figure 5 sensors-23-06184-f005:**
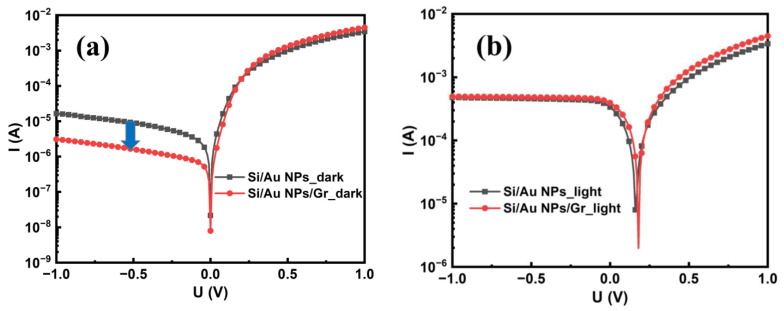
I-V characteristics of textured Si/Au NPs and textured Si/Au NPs/Gr PD device under dark conditions (**a**) and 1319 nm light illumination with incident power of 0.1 W (**b**). The measured EQE at 1100–1600 nm wavelength range of textured Si/Au NPs/Gr PD (**c**).

**Figure 6 sensors-23-06184-f006:**
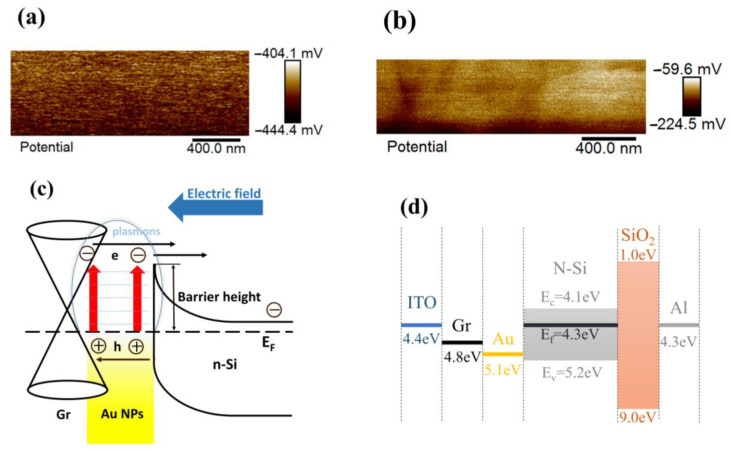
Surface potential of planar Si/Au NPs (**a**) and planar Si/Gr (**b**). The energy band diagram of textured Si/Au NPs/Gr (**c**) and the band alignment of each component in the PD device (**d**).

**Figure 7 sensors-23-06184-f007:**
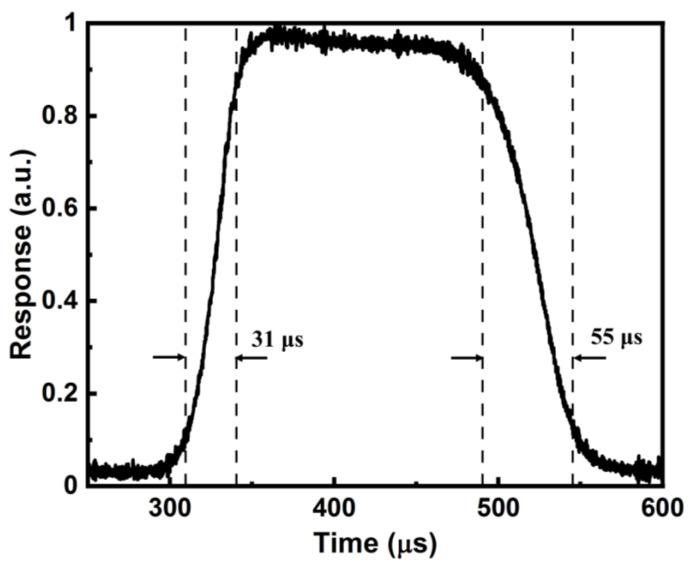
Temporal response of the Si/Au NPs/Gr device at room temperature under 0 V bias.

**Table 1 sensors-23-06184-t001:** Performance parameters of textured Si/Au NPs PD without and with Gr working at room temperature (RT) and low temperature, compared with the PDs in the literature.

PD Device	R (mA/W)	D* (cm × Hz^1/2^/W)	J_dark_ (nA/cm^2^)	Ref.
Si/Au NPs (RT)	3.4	4.1 × 10^10^	22	This work
Si/Au NPs/Gr (RT)	3.9	7.2 × 10^10^	9	
Si/Au NPs/Gr (−60 °C)	3.9	1.5 × 10^11^	2	
Si/Au NPs	5.8	2.3 × 10^10^	200	[24]
Si/Au	1.7	5.1 × 10^9^	356	[41]
Si/monolayer Gr (850 nm)	29	3.9 × 10^11^	17	[42]
Ge/monolayer Gr (1550 nm)	51.8	1.4 × 10^10^	~4 × 10^4^	[43]

## Data Availability

Not applicable.

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
