# Peer review of "A Silicon Sub-Bandgap Near-Infrared Photodetector with High Detectivity Based on Textured Si/Au Nanoparticle Schottky Junctions Covered with Graphene Film"

_sensors, 2023, doi:10.3390/s23136184_

Round 1

Reviewer 1 Report

1- In line 12 the word "developing" should be "develop".

2- In section 2.1. Preparation of textured Si substrate can authors also mention how did they pattern the pyramids? Did they use photolithography?

3-  Can authors elaborate on the advantages of choosing ITO and Al for electrodes over other metal electrodes?

4- Did the authors study the conduction through SiO2? How does the SiO2 thickness effect the conduction

5- In line 89 it should be in-house

6- The first paragraph of section 3.  Results and discussion should be part of the Experiment section as this mostly discuss the structure details, not the results.

7- In Figure 2(b) authors should visually indicate the Au NPs of size 40 nm and 100 nm in the SEM image

The overall writing style and structure are good. Authors can pay attention to repeating the statements.

Reviewer 2 Report

In this manuscript entitled “Silicon sub-bandgap near-infrared photodetector with high detectivity based on textured Si/Au nanoparticles Schottky junctions covered by graphene film”, Dai and coauthors have reported on the fabrication of near-infrared photodetectors by coupling graphene, Au nanostructures, and patterned silicon. Impressively, the graphene/Au/Si heterojunction photodetector exhibits pronounced photoresponse to 1319 nm illumination, which is beyond the limit of intrinsic silicon (<1110 nm). Under a self-powered working mode, a responsivity of 3.9 mA/W and a detectivity of 7.2×1010 Jones are achieved. In addition, the device demonstrates fast response rate with the response/recovery time of 31/55 μs. On the whole, this study has great significance as the near-infrared spectrum is critical for applications such as optical communications. Nevertheless, there seem to be several pivotal issues to be address to meet the standard of Sensors. Therefore, a major revision is recommended. The following comments should be fully taken into account.

1. The interfacial built-in electric field is critical to the device properties of the graphene/Au/Si heterojunction photodetector. Therefore, KPFM should be conducted to quantificationally investigate the band alignment of the graphene/Au/Si heterojunction. In addition, it is claimed that the photoexcited electrons will transport from graphene to Si. However, in Figure 6, graphene is p-type and silicon is n-type. Theoretically, a p-n junction is formed with the built-in electric field pointing from silicon to graphene and photoexcited electrons should transport along the opposite direction. How to explain the contradiction?

2. Light trapping of the textured Si is an important factor contributing to the excellent device properties. Therefore, FDTD simulation should be conducted to explore the light intensity distribution of textured Si as compared to planar silicon to provide a more in-depth insight.

3. The Au NPs are known to possess strong plasmonic effect (e.g., Small 2015, 11, 2392; ACS Nano 2017, 11, 10321; ACS Nano 2018, 12, 8739), which may be critical factors promoting the photosensitivity. However, I can’t find related content in the current version of the manuscript. It should be clearly discussed.

4. In addition to responsivity and detectivity, EQE is also an important performance metric of photodetectors. It should be extracted.  

5. A table summarizing the performance metrics of this device and state-of-the-art self-powered photodetectors operated in the similar spectrum (not only limited to silicon) should be provided to provide clear comparison.

6. Some minor errors. a) Page 1, “to developing” should be “to develop”. b) Page 1, Photodetectors (PDs), which convert visible or infrared light into electrical signals…Actually, photodetectors are not limited to visible or infrared light. Therefore, “visible or infrared” should be deleted. c) Figure 6 & Figure 7, there should be a blank between the physical symbols and the units. The whole manuscript needs to be carefully checked.

Round 2

Reviewer 2 Report

In general, the authors have addressed most of my previous comments. But I still have some concerns:

1. As shown in Figure 6a & b, the surface potential of Si/Au is higher than that of Si/Au NPs/Gr, which in fact indicates that the Fermi level of graphene is lower than that of Au (e.g., Adv. Opt. Mater. 2019, 7, 1900815; ACS Nano 2023, 17, 6534–6544). This contradicts the explanation on page 8. The results should be carefully checked. It is suggested that the authors should perform KPFM measurement on the interface of Si/Au and Si/Au/Gr. The measurement on the same piece of sample can be more convincing.

2. It is suggested that the results in point 2 of the response can not demonstrate the light trapping effect of textured silicon, as no obvious increasement in absorption occurs in the visible range.

3. As for the sentence of “Photodetectors (PDs), which convert light into electrical signals, play a crucial role in various fields such as data transmission, night vision imaging, biomedical testing, and safety surveillance”, the state-of-the-art advancements (e.g., Mater. Futures 10.1088/2752-5724/acdd87; Nanomaterials 2022, 12, 3775; Small Methods 2022, 6, 2200583) with specific applications are recommended to broaden the readers’ horizons and increasing credibility.
